# Evaluation of Non-Vector Transmission of Usutu Virus in Domestic Canaries (*Serinus canaria*)

**DOI:** 10.3390/v16010079

**Published:** 2024-01-03

**Authors:** Aude Blanquer, Felipe Rivas, Mazarine Gérardy, Michaël Sarlet, Nassim Moula, Ute Ziegler, Martin H. Groschup, Daniel Desmecht, Thomas Marichal, Mutien Garigliany

**Affiliations:** 1Fundamental and Applied Research for Animals & Health (FARAH), Laboratory of Pathology, Faculty of Veterinary Medicine, University of Liège, Sart Tilman B43, B-4000 Liège, Belgium; aude.blanquer@uliege.be (A.B.); jfarivas@uliege.be (F.R.); mazarine.gerardy@uliege.be (M.G.); michael.sarlet@uliege.be (M.S.); daniel.desmecht@uliege.be (D.D.); 2Animal Resources Veterinary Management Department, Faculty of Veterinary Medicine, GIGA Research (AFT), Sart Tilman B23B, B-4000 Liège, Belgium; nassim.moula@uliege.be; 3Friedrich-Loeffler-Institut, Institute for Novel and Emerging Infectious Diseases, Südufer 10, 17493 Greifswald-Insel Riems, Germany; ute.ziegler@fli.de (U.Z.); martin.groschup@fli.de (M.H.G.); 4Laboratory of Immunophysiology, GIGA Institute, University of Liège, B-4000 Liège, Belgium; t.marichal@uliege.be; 5Faculty of Veterinary Medicine, University of Liège, Sart Tilman B42, B-4000 Liège, Belgium

**Keywords:** *Serinus canaria*, usutu virus, infection, horizontal transmission

## Abstract

Usutu virus (USUV) is a flavivirus transmitted to avian species through mosquito bites that causes mass mortalities in wild and captive bird populations. However, several cases of positive dead birds have been recorded during the winter, a vector-free period. To explain how USUV “overwinters”, the main hypothesis is bird-to-bird transmission, as shown for the closely related West Nile virus. To address this question, we experimentally challenged canaries with intranasal inoculation of USUV, which led to systemic dissemination of the virus, provided the inoculated dose was sufficient (>10^2^ TCID_50_). We also highlighted the oronasal excretion of infectious viral particles in infected birds. Next, we co-housed infected birds with naive sentinels, to determine whether onward transmission could be reproduced experimentally. We failed to detect such transmission but demonstrated horizontal transmission by transferring sputum from an infected to a naive canary. In addition, we evaluated the cellular tropism of respiratory mucosa to USUV in vitro using a canary tracheal explant and observed only limited evidence of viral replication. Further research is then needed to assess if and how comparable bird-to-bird transmission occurs in the wild.

## 1. Introduction

Over the last few years, the emergence of several mosquito-borne flaviviruses has been observed all around the world. These viruses have the capacity to affect a wide variety of species, including mammals, birds, and reptiles, with considerable variations in virulence. Usutu virus (USUV) is closely related to West Nile virus (WNV), and both are classified in the Japanese encephalitis virus (JEV) serogroup, in the *Flavivirus* genus of the *Flaviviridae* family [1]. The main hosts of this virus are avian species, in which it is transmitted by ornithophilic mosquitoes, mostly *Culex pipiens*. USUV has been responsible for several epornitics in Europe [2,3,4,5,6], often resulting in an important die-off, with a major impact on wild and captive populations of birds [7], especially in common blackbirds (*Turdus merula*) [8,9]. USUV is transmitted to a wide variety of avian hosts, which may result in different clinical signs depending on the species infected, ranging from mild symptomatology for resistant species to severe multisystemic disease with a high mortality rate in susceptible species [5]. 

Although mosquito bites represent the main route of transmission, we cannot exclude that other routes are involved in avian species. Indeed, evidence of infection caused by WNV and USUV has been observed during the vector-free period, which occurs mostly during the winter [10,11,12,13]. The ability of a vector-borne virus to pass through the winter, without any competent insect vector to allow its transmission and maintenance is called “overwintering”. Although some cases of chronic disease with persistent viremia after WNV infection have been described in susceptible bird species like the crow [14] or house sparrow [15], direct transmission in birds remains the most likely hypothesis. Mosquito bites seem unlikely to be the sole cause of countless deaths in bird populations observed during outbreaks. Moreover, bird-to-bird transmission has been highlighted for WNV in several bird species [16,17,18]. Furthermore, different routes of shedding have been investigated, and viral particles were found in feces, feathers, and oral and cloacal swabs of naturally and experimentally WNV- [15,18] and USUV-infected birds [19,20]. Even if nest sharing in a big community of birds seems auspicious for fecal–oral transmission, no WNV RNA was detected in the samples of feces collected from three crow roosts despite evidence of WNV circulation [14]. Another route of infection seems more likely, namely the oro-nasal entry route. Human nasal epithelial cells appear to be highly permissive to several flaviviruses, including WNV and USUV [21]. It has also been shown that intranasal inoculation of WNV and USUV in mice could lead to systemic dissemination of the virus [22,23]. Similar observations have been made for Langat virus (LGTV) and tick-borne encephalitis virus (TBEV) in mice, showing horizontal transmission of the virus, through direct contact and a contaminated environment, but not through aerosols [24]. Furthermore, even if USUV is not a respiratory virus, high viral titers were found in the lungs of infected birds after a parenteral infection for several species [19,20], which suggests that this organ could also be a site of viral replication. 

Based on all these data, we wondered if direct transmission of USUV, most likely via the oro-nasal route, might be partly responsible for the overwintering phenomenon and contribute to the massive die-off events observed in wild and captive bird populations during outbreaks in Central and Western Europe. 

## 2. Materials and Methods

### 2.1. Virus

For the challenge, the Usutu strain USU-BE-Seraing/2017 (GenBank: MK230892, lineage Europe 3) was used. The strain was isolated from the organs of a European blackbird found dead during an avian outbreak in 2017 in Belgium [25]. The virus was amplified in African green monkey Vero cells (ATCC CRL-1586) and titrated using the 50% tissue culture infective dose (TCID_50_) technique in Dulbecco’s minimum essential medium (DMEM) (Gibco^®^, London, UK), supplemented with 2% of heat-inactivated fetal bovine serum (FBS) (Biowest^®^, Nuaillé, France) and 1% of penicillin/streptomycin (Gibco^®®^ Antibiotic-antimycotic 15240-062, UK).

### 2.2. Canary Experiments

All canaries (*Serinus canaria*) involved in our study were six-month-old males, provided by a breeding facility (Animalerie Smets, Oupeye; certification number: HK51603061) and installed into the biosafety level 2 (BSL2) experimental animal facility of the Department of Pathology at the Faculty of Veterinary Medicine in Liège, Belgium. They were marked by numbered colored leg rings and housed in cages with water and grains ad libitum. A blood sampling was performed under anesthesia with 5% isoflurane inhalation one week before inoculation in order to test for the presence of anti-WNV/USUV antibodies prior to the infection. All injections/inoculations were performed under anesthesia with 5% isoflurane inhalation. During the experiments, canaries were monitored twice per day and weighed daily for 15 days post-infection. Neurological symptoms or a weight loss of over 20% of their initial weight were fixed as humane endpoints requiring euthanasia. The animal welfare and all procedures performed on the canaries were approved and supervised by the Committee for Ethics in Animal Experimentation of the University of Liege, Belgium (Identification code: 21-2363, date of approval: 14 July 2021).

#### 2.2.1. Comparison of the Intranasal and Intradermal Inoculation Routes of USUV in Canaries

A first group of six canaries was enrolled and received 10^6^ TCID_50_ of USUV by the intranasal route, dispersed in 25 µL of DMEM (12.5 µL per nostril). In parallel, as parenteral control, six canaries received 10^6^ TCID_50_ of USUV by the intradermal route (in the skin of the chest), dispersed in 25 µL of DMEM. Given the thinness of the canary dermis, a 30 G insulin syringe (BD Micro-FineTM + Demi 30G × 8 mm U-100 0.3 mL; BD, Wokingham, UK) was used for the injection. Moreover, to optimize the intradermal route, each inoculum was injected into two separate sites, with 12.5 µL in each site. A second experiment was carried out in order to determine the dose–effect response with different viral concentrations after intranasal inoculation. To that end, two groups of six canaries received 10^4^ TCID_50_ or 10^2^ TCID_50_, respectively, of USUV by the intranasal route, as previously described.

#### 2.2.2. Evaluation of the Horizontal Transmission of USUV in Co-Housed Sentinels

The infected group was composed of six canaries having received 10^6^ TCID_50_ of USUV, intradermally or intranasally, following the protocol described above. The sentinel group was composed of six naive canaries. Both groups, infected and sentinels, were co-housed and shared the same feeding and drinking places for 15 days after the infection. 

#### 2.2.3. Evaluation of “Forced” Transmission of USUV in Canaries

We assessed the possibility of indirect transmission in order to mimic what happens during chick feeding in the wild, or forced feeding in rehabilitation centers. We intradermally infected a group of four canaries with 10^6^ TCID_50_ of USUV, as described above. An oral swab was then collected at the peak of excretion (previously determined) from these canaries, and directly inoculated into the oral cavity of a group of four naive canaries. Each naive canary received the swab of an infected canary, and the couples created were identified using colored leg rings. The parameter that was evaluated here was the occurrence or not of a seroconversion (protocol detailed below) in the canaries having received the infected swab.

### 2.3. Sample Collection

As one of the main goals of the study was the evaluation of possible non-vector transmission of USUV, several excretion routes were assessed in the first experiment. Thus, fresh droppings were harvested every day between days 1 and 6 post-infection. Samples of water in drinking places were also collected. Finally, swabs in the oral cavity were performed on all canaries, both infected and sentinels, on days 2, 4, and 6 post-infection. All these samples were mixed with 500 µL of DMEM and centrifuged at 1400× *g* for 5 min, then filtrated on a 0.2 µm membrane (Acrodisc^®^ 32 mm Syringe Filter with 0.2 µm Supor^®^ Membrane; PALL^®^, Port Washington, NY, USA). In the case of euthanasia of a canary when a critical point was reached, a necropsy was performed and 25 ± 1 mg of the spleen, liver, brain, lungs, kidney, eye, and heart were collected and stored at −80 °C for PCR analysis. Other portions of these organs, as well as the skin and trachea, were fixed in paraformaldehyde 4% for histological and immunohistochemical evaluations. At 15 days post-infection (dpi), all surviving canaries were euthanized, and blood was collected. Serum and clot were separated by centrifugation at 1400× *g* for 10 min and the serum was stored at −20 °C for antibody detection analysis.

### 2.4. Virus Isolation and Titration

USUV isolation was performed as in [20]. Briefly, filtrates of samples of droppings, water, and swabs were dropped on Vero cells in 6-well plates with complete DMEM medium and incubated at 37 °C 5% CO_2_. Plates were checked daily for the occurrence of cytopathic effects. Three blind passages on fresh Vero cell culture were performed after 5 days of incubation. After the last passage, supernatants were tested by RT-qPCR to confirm the presence of USUV observed on the basis of visualization of cytopathic effects. However, as several passages on Vero cells had been performed, the quantification of the initial number of viral particles was no longer possible. The titration of the oral swabs was therefore carried out using *Aedes albopictus* clone C6/36 cells (ATCC CRL-1660), whose permissivity to flaviviruses has been demonstrated in the literature [26,27,28]. Briefly, filtrates of samples of oral swabs were titrated using the TCID_50_ technique in Roswell Park Memorial Institute medium (RPMI 1640) with L-glutamine and 25 mM HEPES (Biowest^®^, France), supplemented with 2% of heat-inactivated fetal bovine serum (FBS) (Biowest^®^, France), 1% of penicillin/streptomycin (Gibco^®^ Antibiotic-antimycotic 15240-062, UK), 1% of sodium pyruvate (Gibco^®^, UK), and 1% of MEM non-essential amino acid solution (Lonza^®^, Durham, NC, USA), and the plates were incubated at 28 °C 5% CO_2_ for 5 days. 

### 2.5. Viral Detection by RT-qPCR

Total RNA was extracted from collected organs, swabs, droppings, and cell culture supernatants using the TANBead^®^ Nucleic Acid Extraction Kit OptiPure Viral Auto Tube (Taiwan Advanced Nanotech^®^, Ref. W665S66) with the extraction robot Maelstrom 9600 (Taiwan Advance Nanotech^®^, Taoyuan, Taiwan). The total RNA of each sample was previously standardized based on quantification using Isogen Life ScienceTM’s NanoDrop^®^ Spectrophotometer (ND-1000). USUV RNA was detected and measured by absolute quantification using a reverse transcriptase quantitative polymerase chain reaction (RT-qPCR), according to [5]. Briefly, the following primers were used for the RT-qPCR: forward 5′-CGTTCTCGACTTTGACTA-3′; reverse 5′-GCTAGTAGTAGTTCTTATGGA; probe: 5′-ACCGTCACAATCACTGAAGCAT-3′. The Luna^®^ Universal One-Step RT-qPCR Kit (New England Biolabs Inc., Ipswich, MA, USA) was used, under the following conditions: retro transcription for 20 min (minute) at 45 °C; inactivation and initial denaturation at 95 °C for 10 min; then 40 cycles of amplification: 95 °C for 15 s (second), 48 °C for 20 s, 72 °C for 60 s; and a final extension at 72 °C for 1 min. The number of viral RNA copies was then calculated by absolute quantification using a standard curve, as described previously [19]. 

### 2.6. Histopathology and Immunohistochemistry

After paraffin embedding, tissue samples were sectioned (5 µm thick) and stained with hematoxylin and eosin. To test for the presence of viral antigens, slides were processed for immunohistochemistry (IHC) using a polyclonal rabbit anti-USUV antibody, as described in [29], with some modifications. Briefly, sections were deparaffinized and rehydrated, and an antigen retrieval step was performed using distilled water in a microwave, 3 × 5 min at 600 W. Then, slides were incubated for 10 min with H_2_O_2_ 0.35% to block the endogenous peroxidases. Unspecific reactions were blocked with Animal-Free Blocker^®^ (Vector Laboratories, Burlingame, CA, USA) for 25 min at room temperature. The anti-USUV antibody was used at a 1:2000 dilution in Tween 20/phosphate-buffered saline (PBS) (1:5 dilution) for 1 h at 37 °C. The negative control was incubated with Tween 20/PBS. An anti-rabbit secondary antibody (EnVision+ System-HRP Labelled Polymer Anti-Rabbit; Dako^®^, Carpinteria, CA, USA) was used for 30 min at room temperature. Then, a chromogen (AEC+ High Sensitivity Substrate Chromogen; Dako^®^, Carpinteria, CA, USA) was used according to the manufacturer’s instructions, and a counterstaining was performed with Gill’s hematoxylin.

### 2.7. USUV Antibodies Detection

Serum samples collected prior to infection and at the end of both experiments were screened for antibodies against USUV with a competitive ELISA kit (ID Screen^®^ West Nile Competition Multi-species, Grabels, France). This kit is not designed to work with USUV in particular but it contains the WNV envelope protein, allowing the fixation of immunoglobulins M and G on the common epitope of viruses from the Japanese encephalitis virus serocomplex [19,30].

### 2.8. Tracheal Explants

In order to determine which cell types are involved in the viral replication in the respiratory tract, an ex vivo infection of the canary tracheal explants was performed. To achieve this goal, we followed the protocol described in [31,32], with some modifications. Briefly, tracheas from two canaries were carefully dissected, and each was sectioned into three parts, to work in triplicates. Each segment was placed in an individual well in a 24-well plate in 1 mL of complete DMEM medium, as previously described. First, a kinetic study of the viral replication was performed. After infection with 10^5^ TCID_50_/well of USUV and incubation for 1 h 30 min at 37 °C 5% CO_2_, the inoculum was removed, and fresh medium was added. Culture supernatants were harvested at 0, 24, and 48 h post-infection and the samples were analyzed for USUV RNA by RT-qPCR. In parallel, in order to visualize which type(s) of cells contain viral antigens, triplicates of tracheal segments were infected as described previously, but with 10^6^ TCID_50_/well. The infection was stopped after 12 h, and segments were fixed in paraformaldehyde 4% for 2 h and then embedded in paraffin wax for IHC evaluation of viral antigens.

### 2.9. Statistical Analyses

All analyses were performed using SAS (version 9.3) and significance levels were set at 5%. All variables were checked for normality assumption. Survival curves were plotted and compared using the log-rank and the Gehan–Breslow–Wilcoxon tests in GraphPad Software (version 9, La Jolla, CA, USA).

The Kruskall–Wallis test was used to study the effect of time (T0 h pi, T24 h pi, and T48 h pi) on Viral RNA loads. The Wilcoxon–Mann–Whitney test was used to study the effect of virus isolation (negative, positive) on Ct value. The Fisher test was used to compare the rate of seroconversion between infected and sentinel groups. 

## 3. Results

### 3.1. Susceptibility of Canaries to the Intranasal Inoculation of USUV

Although USUV is classically transmitted to avian species via the bite of a mosquito, other routes of transmission are suspected to explain the outbreaks observed during vector-free periods. As bird-to-bird transmission appears as the likely hypothesis, we addressed this question by challenging canaries with an intranasal inoculation of USUV and compared the outcome with the intradermal route. 

After intranasal inoculation (i.n.) of USUV with 10^6^ TCID_50_/canary, one canary started to show clinical signs, such as ruffled feathers, lethargy, isolation, and depression at 8 dpi. Because of the deterioration of its general condition, this bird was euthanized and a substantial loss in the initial body weight of 15.3% was recorded. The other infected canaries did not show any symptoms and all survived (Figure 1).

In parallel, the intradermal inoculation (i.d.) of USUV led to comparable symptoms in two canaries, i.e., depression and lethargy between 5 and 6 dpi, and the birds were euthanized at days 6 and 7 pi, respectively (Figure 1). A loss in the initial body weight of 10.4% and 14.7%, respectively, was recorded for these birds.

Interestingly, contrary to mammals, the infected canaries did not show evidence of typical neurological symptoms of encephalitis but rather a sudden and severe depression, leading to death the day after. Moreover, canaries succumbed to the infection about a week after the infection, regardless of the inoculation route. 

Based on previous data showing evidence of oro-nasal excretion of USUV, we assessed which might be the minimal dose needed in intranasal inoculation to generate a clinical disease in canaries. We then noticed the appearance of typical clinical signs in one canary infected with 10^4^ TCID_50_ after 9 dpi (Figure 2). The loss in the initial body weight of this canary was 19.8%. However, in the 10^2^ TCID_50_ group, no canary showed any symptoms and there were no mortalities. 

At the end of the experiment, i.e., 15 dpi, the serological analyses showed that after intradermal inoculation of USUV, all surviving birds were positive for the presence of USUV antibodies. Concerning the groups intranasally infected with 10^6^ TCID_50_ and 10^4^ TCID_50_, 50% (the bird that succumbed to the infection and two birds out of the five survivors) and 33% (the bird that succumbed to the infection and one bird out of the five survivors), respectively, showed evidence of seroconversion. All canaries infected with 10^2^ TCID_50_ were negative. 

All dead canaries showed high amounts of USUV RNA in their organs as determined by RT-qPCR, with variations between organs and time of death, as shown in Table 1.

### 3.2. Histopathological Lesions and Cellular Tropism Are Independent of the Route of Inoculation of USUV

During the necropsy of dead canaries, the main gross lesions were a pallor of the liver and splenomegaly. Microscopic findings in these organs consisted of a multifocal moderate mononuclear inflammation and necrosis in the liver and a slight lymphoid depletion in the spleen. After intradermal inoculation, lymphoplasmacytic and histiocytic infiltration in the dermis and adjacent muscle was observed at the inoculation site (Figure 3A,B). A similar infiltrate was present in the mucosa/submucosa of the trachea and nasal turbinates of dead canaries after intranasal inoculation (Figure 3C,D). No specific pathological findings were made in the brain, heart, and lungs.

All lethally infected birds showed evidence of USUV antigen immunolabeled cells randomly distributed in the spleen and liver at the time of death. Large numbers of macrophages were positive in the liver, with numerous mononuclear leucocytic positive cells in the spleen (Figure 4A,B) of i.d.- and i.n.-infected canaries dead at 6, 7, and 8 dpi, respectively. Concerning the trachea, positive mononuclear leucocytic cells were observed in the lamina propria (Figure 4C), but only in the case of canaries dead after intranasal inoculation at 8 dpi. Respiratory epithelial cells did not show any evidence of staining, just as the lung, heart, brain, and skin at the time of death. 

### 3.3. The Avian Airway Mucosa Is Only Weakly Permissive to USUV

The intranasal inoculation of USUV leads to a multisystemic disease very similar to that observed after parenteral inoculation. Since human respiratory epithelial cells are permissive to USUV [21], we wondered about the cellular tropism of USUV for the airway mucosa. To address this question, we infected tracheal explants of canaries ex vivo. The kinetic and quantitative analysis of the concentration of viral RNA in the culture supernatant at 24 and 48 h post-infection showed a slight viral amplification by tracheal cells (Figure 5), but not statistically significant (*p* = 0.11). A few respiratory epithelial cells appeared slightly positive for viral antigens by immunochemistry (Figure 6A). However, strong USUV labeling was present in some foci of cells in the adipose tissue, surrounding the trachea (Figure 6B). A more accurate characterization of these cells is necessary to determine their exact nature. Muscular cells next to these cells also showed positive labeling, but less intense (Figure 6A,B).

### 3.4. Infected Canaries Shed Infectious USUV Particles

To assess the occurrence of viral excretion by infected birds, we collected samples from the environment (droppings and water), but also from the oral cavities of infected canaries. All the swab samples collected on days 2, 4, and 6 were analyzed by RT-qPCR and the results were compared with those obtained after virus isolation on Vero cells. First, it appeared that there was a link between the presence of cytopathic effects in cell cultures after three blind passages and the detection of USUV RNA by RT-qPCR. Positive wells corresponded to oral swabs collected from lethally infected birds on days 2, 4, and 6, and, for survivors, on day 4 only. However, among survivors, we noticed that 100% of intradermally infected canaries showed a shedding of infectious particles at day 4, whereas this was only the case for 50% of the intranasally infected group with 10^6^ TCID_50_ and only 17% with 10^4^ TCID_50_. In addition, because of the high permissivity of the C6/36 cells for USUV, we performed a viral titration of oral swabs collected from intradermally infected birds. We observed viral titers from 10^2^ to 10^3^ TCID_50_/swab on day 4 in 100% of individuals and only in 33% on day 6. Moreover, the analysis of oral swabs presented two different scenarios. For canaries which succumbed to the infection (i.d. or i.n.), high viral RNA levels from 6.06 ± 0.06 to 5.37 ± 0.07 log10 viral RNA copies (VRC)/swab (RT-qPCR cycle thresholds (Ct) values from 23.2 to 25.6) were observed on swabs collected from day 2 to day 6 pi, while for all other canaries, viral RNA loads were higher on days 2 and 6 with values from 3.84 ± 0.04 to 1.77 ± 0.06 VRC/swab (Ct values from 30.7 to 37.6) than on day 4 with values from 6.30 ± 0.06 to 4.07 ± 0.07 VRC/swab (Ct values from 22.5 to 29.9). Thus, the peak of the viral excretion phase seems to be around day 4 pi in all infected animals. A meaningful link also appeared between the mortality and the possibility of isolating infectious virus on days 2, 4, and 6 pi (*p* < 0.0001). Thus, an association exists between the intensity of oro-nasal excretion and the survival of the animal.

Interestingly, it appears that the threshold needed to isolate infectious viruses from oral swabs corresponds to a Ct value inferior to 30.0. This observation was confirmed by statistical analyses, as shown through the box plot in Figure 7. 

All detailed results for each animal are presented in the Appendix A.

Regarding the collected droppings, only low amounts of viral RNA were found. For the intradermally infected group, a mean of 2.16 ± 0.04 log10 viral RNA copies (VRC)/50 mg at day 1 and 1.89 ± 0.03 log10 VRC/50 mg at day 2 was recorded. For the group infected via the intranasal route, 2.28 ± 0.04 log10 VRC/50 mg at day 1 and 1.89 ± 0.04 log10 VRC/50 mg at day 2 were observed. No longer viral excretion was observed in collected droppings from days 3 to 6. However, no infectious virus isolation was possible after the cultivation of droppings samples. In addition, all collected water samples were negative.

### 3.5. Evidence of Horizontal Transmission of USUV Infection to Naive Canaries

As we showed the oral excretion of infectious viral particles by infected canaries, we wondered about the possibility of direct or indirect transmission of USUV in a bird population. To address this question, infected canaries (i.n. or i.d.) and naive sentinels were co-housed and shared the same environment for 15 dpi (Figure 8). Among sentinels, neither clinical signs nor mortalities were recorded. An analysis of oral swabs showed that some viral RNA was present in the oro-nasal cavity at 4 dpi in 50% of sentinels (Ct range: 37.8 to 38.2) in contact with i.d.-infected canaries, but none among sentinels in contact with the i.n.-infected group. Interestingly though, 100% and 83% of the sentinels, respectively, in contact with i.n.-infected and i.d.-infected birds had viral RNA in their oral cavity at 6 dpi (i.n. Ct range: 35.0 to 37.7; i.d. Ct range: 37.9 to 38.1). Despite positive PCR results, no infectious viral particles were isolated on Vero cell cultures.

Spleen and liver samples of the sentinels were PCR-negative for viral RNA, and no evidence of seroconversion was recorded after 15 days of co-housing (Figure 8).

The study of the viral shedding by infected birds highlighted a peak of oral viral excretion at 4 dpi. To address the possibility of indirect horizontal transmission, we collected oral swabs from four infected canaries at 4 dpi and used these swabs to “inoculate” four naive canaries by the oral route (Figure 9). Among these birds, neither clinical signs nor mortalities were recorded. However, evidence of seroconversion was recorded in one canary out of four 15 days after the inoculation (Figure 9).

## 4. Discussion

The emerging mosquito-borne flavivirus USUV has been responsible for massive mortalities in wild or captive bird populations [6,8,9]. Cases of USUV- or WNV-positive dead birds have also been reported during the winter, a vector-free period [10,11]. Therefore, it cannot be excluded that other routes of transmission than mosquito bites are involved, at least in avian species. The most likely hypothesis to explain such winter outbreaks is bird-to-bird transmission, which has been highlighted for WNV in several bird species [16,17,18]. The aim of this study was to investigate the importance of horizontal transmission of USUV in birds.

In this study, we assessed two particular routes of USUV inoculation in domestic canaries, i.e., intranasal and intradermal (as a proxy for mosquito bites). We showed that the systemic dissemination of the virus was independent of these inoculation routes. Interestingly, the viral amplification in the body of intranasally infected birds was strongly influenced by the inoculation dose, i.e., a 10^6^ TCID_50_ USUV inoculum per canary was by factors more efficient than 10^4^ or 10^2^ TCID_50_. Co-housing of sentinel birds with either intranasally or intradermally infected animals, did not reveal any horizontal transmission, albeit scarce virus genomes could be revealed in the throat swabs of the sentinel birds in contact with intradermally infected canaries. This motivated us to conduct an experiment on transmission from intradermally infected canaries to sentinel birds. Interestingly, one out of four sentinel canaries developed USUV antibodies after such exposure, which is a strong indication that clinically infected animals can infect their fellow sentinels via their sputum, i.e., horizontally. 

In order to assess the intranasal and intradermal inoculations of USUV as possible infection routes, we used an experimental model already validated in the laboratory, the domestic canary [19]. The choice of the canary species was made with regard to its belonging to the *Passeriformes* order, whose members were found to be highly susceptible to USUV infection in the wild [2,33]. Moreover, it has been proven that the inoculation route could have an impact on viral dissemination in mice [22,34]. In addition, the intradermal route best mimics the inoculation by a mosquito bite [35]. We showed that both routes lead to a similar systemic dissemination of the virus, and can result in the death of the canary. It thus appears that the local viral replication in the respiratory airways is sufficient to generate a comparable disease in canaries. Interestingly, we observed a very similar local host reaction at the inoculation site (skin or nasal turbinates) for both routes, with a prominent lymphoplasmacytic and histiocytic response. This strikingly similar response whatever the infection route suggests that similar mechanisms are involved in the host immune response. Indeed, after the initial amplification of the virus, a massive inflow of permissive leucocytes attracted to the site of viral inoculation seems to be the key to flavivirus dissemination through the organism [36,37,38]. Incoming leukocytes could thus be the main actors of the pathogenesis of mosquito-borne flaviviruses and might explain the difference in susceptibility between hosts [39]. Further studies are necessary to elucidate this point.

High viral RNA loads in the spleen, liver, and lungs, and large numbers of antigen-positive cells especially in the spleen and liver were found in lethally infected birds. The lack of antigen-positive cells in the brain, liver, and lungs, in spite of the detection of significant viral RNA loads, might reflect the blood distribution of viral RNA (either free or in virions) in these organs rather than local replication. Indeed, we previously showed that canaries lethally infected by USUV have very high RNAemia levels during the course of their infection [19]. It thus appears from these observations that the key event in the viral pathogenesis in avian hosts is the systemic spread of the virus, whatever the inoculation route. Strikingly, the brain appeared as the organ, among those we sampled, containing the lowest amounts of viral RNA. This is consistent with the lack of detection of antigen-positive cells by IHC, but also with the absence of neurological symptoms during our experiments. USUV infection in birds appears more as a systemic than a pure neurotropic disease, in opposition to what has been described in mammals [22,40,41]. These data are consistent with those obtained after intraperitoneal inoculation in a previous study in the laboratory [19], but also with the observations made in canaries after subcutaneous WNV infection [42]. Even if WNV appeared more virulent than USUV, with a higher mortality rate, the antigen detection by IHC showed similar results, with strongly positive cells mostly found in the liver and spleen. This therefore suggests a marked tropism of flaviviruses for these organs in birds, making them important sites of viral replication. Further, IHC analysis revealed that the most consistently positive cells were likely of histiocytic type, their exact nature remaining to be determined. 

However, we noticed that the mortality rate in our experiment was lower than previously observed by Benzarti et al. [19]. We also noticed that the neurotropism of USUV was quite less pronounced here than in the case of naturally infected birds. Indeed, neurological symptoms were observed in wild birds, confirmed by multifocal neuronal necrosis in the brain [4,33]. These differences between naturally and experimentally infected birds could be explained by the several passages in Vero cells of the USUV stock used. Moreover, even though we tried to mimic a natural infection, we could not reproduce all the biological parameters of mosquito-borne transmission. Indeed, the critical role of components of mosquito saliva has been described in the literature on flaviviruses [43,44], including WNV [45,46]. Mosquito saliva enhances WNV replication in vitro [47], but also in vivo in mammals and birds [46,48]. In addition, the amount of USUV infective particles inoculated by an infected mosquito is unknown. For WNV, it was estimated to vary from 10^3.4^ to 10^6.1^ PFU, depending on the mosquito species [49]. These data and the study by Benzarti et al. [19] are consistent with what we observed in the dose–effect study, where we highlighted that the minimal infectious dose for an intranasal infection was superior to 10^2^ TCID_50_/canary. 

In addition, as we showed that the intranasal inoculation of USUV led to systemic dissemination of the virus, we wondered which were the resident cell type(s) responsible for the initial amplification of the virus inoculated in the nasal cavities of the canary. Indeed, human nasal epithelial cells were shown to be permissive to several flaviviruses (including USUV) in vitro [21]. It was thus tempting to suggest that avian respiratory epithelial cells might also be a site of viral replication after intranasal inoculation. We addressed this question using ex vivo tracheal explants from naive canaries, as previously described in mice [31] and chicken [32]. Indeed, trachea and nasal turbinates share a similar ciliated epithelium in terms of cellular composition, except for the olfactory epithelium [50,51] and the local immune system [52,53]. Moreover, using explants has the advantage of providing the entire in situ cellular environment but without the intervention of incoming leucocytes. However, in our hands, viral amplification appeared, at best, very limited in the kinetic study, and only a low number of antigen-positive epithelial cells was observed. Consequently, canary respiratory epithelial cells did not seem to be as permissive to USUV as their human counterparts. Nevertheless, another hypothesis could be the role played by nervous system cells, such as neurons, which are also present in the olfactory epithelium, and in which USUV replication has been demonstrated for the central and peripheral nervous system of birds [19] and mice [40]. Further studies are thus necessary to identify the cell type(s) responsible for the initial replication of USUV after an intranasal inoculation.

As an oro-nasal entry route appears as the most likely hypothesis for bird-to-bird transmission of USUV, oral and fecal shedding was monitored in infected canaries. USUV shedding in droppings and feathers had already been proved by our team and others [19,20] but, to our knowledge, an oral excretion of USUV in infected birds had not been assessed yet. Such oral excretion has been shown for WNV in geese [18] but also in canaries [42]. This last study showed that significant amounts of infectious viral particles were excreted by infected birds. The proportion of canaries shedding an infectious virus was dependent on the infectious dose used, with variations in the peak of excretion occurring between 2 and 4 dpi [42]. In our experiment, we highlighted that all birds eventually seroconverted excreted infectious particles at 4 dpi in their oral cavity. For birds that succumbed to the infection, we also isolated infectious viruses at 2 to 6 dpi. We then wondered if the Ct-value could be a good predictor of the contagiousness of USUV, as it was used in human medicine during the SARS-CoV-2 pandemic to assess if a convalescent patient could be discharged from the infectious diseases ward [54,55]. Interestingly, we showed that a Ct-value, determined by RT-qPCR, inferior to 30.0 in oral swabs in our conditions was a good predictor of the possibility of isolating the infectious virus. In parallel, we also monitored the fecal excretion of USUV by infected canaries, showing the presence of low amounts of viral RNA, albeit not infectious. This last observation is in line with what was reported for crow roots with WNV [14]. 

Taking into account the oro-nasal shedding of infectious viral particles and the susceptibility of canaries to the intranasal inoculation of USUV, a direct transmission by this route in avian species is strongly suspected. This bird-to-bird transmission was assessed for WNV for several bird species, especially of the *Passeriformes* order, both in natural [11,14,16] and experimental conditions [17,18]. The aim was to explain the phenomenon of overwintering. All these experiments led to evidence of transmission of WNV in sentinels which were co-housed with infected birds, such as mortalities and/or seroconversion. To experimentally assess this bird-to-bird transmission of USUV, uninfected sentinels shared the same cage as infected canaries (i.n. or i.d.), and, surprisingly, we failed to show such transmission in our experimental conditions. Indeed, all the sentinels survived and there was no evidence of seroconversion. However, we demonstrated a strong dose effect after i.n. inoculation of USUV on the seroconversion rate of canaries. Relatively high doses of USUV were necessary to observe seroconversion in some birds (all birds were negative at 10^2^ TCID_50_/canary, with a seroconversion rate of 33% at 10^4^ TCID_50_/canary). In our experiment, it is likely that the dose transmitted to sentinel birds was noticeably inferior to 10^6^ TCID_50_/canary, necessary to observe a high seroconversion rate. Indeed, viral titers on C6/36 cells of oral swabs from i.d.-infected canaries at day 4 pi ranged from 10^2^ to 10^3^ TCID_50_/swab. Interestingly though, we noticed low amounts of USUV RNA in the oral swabs of the sentinel birds by RT-qPCR. These results suggest that a contact transmission of viral RNA happened between the canaries, but these contacts seemed insufficient to allow a significant transfer of infectious particles to the sentinels. In addition, it is likely that only limited contact occurred between adult canaries during our experiment. Indeed, there are only a few social interactions between adult birds in captivity, contrary to mammals such as mice [56,57], which could additionally explain the lack of transmission. Further, the behavior of adult birds in experimental conditions is quite different from what happens in the wild. Indeed, we created “artificial” groups of male canaries in an environment where enrichment is restricted, limiting the expression of their instinctive habits [56]. In addition, data from the literature suggest that horizontal transmission of flaviviruses is possible, including for the closely related WNV [17,18], through direct contact with infected birds, but also possibly in more specific situations, such as, for example, chicks feeding, picking, or scavenging [10,18]. Forced feeding in rehabilitation centers might also be a favorable situation for the transmission of such viruses. We thus tried to reproduce these specific situations, by inoculating a naive canary with an oral swab collected from an infected canary at 4 dpi. Interestingly, we recorded evidence of seroconversion in one canary out of four after 15 days post-inoculation. It thus appears that horizontal transmission of USUV between birds is possible but under specific conditions, such as sufficient excreted dose and close contact between birds during the peak of the oral excretion phase.

## 5. Conclusions

In summary, evidence of an oral excretion of USUV by experimentally infected canaries leads us to highlight the possibility of horizontal transmission of the virus. Although this transmission only occurred under specific conditions experimentally, it could play a role in a natural environment during the vector-free period. Further studies are thus needed to identify the key factors responsible for bird-to-bird transmission of flaviviruses in the wild.

## Figures and Tables

**Figure 1 viruses-16-00079-f001:**
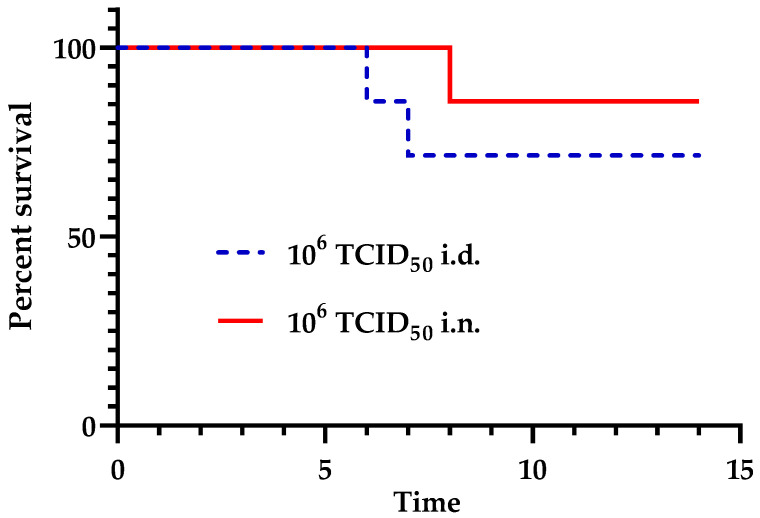
Kaplan–Meier survival curves of canaries infected with USUV via the intradermal or intranasal route with 10^6^ TCID_50_/canary (*n* = 6). The survival curves did not differ statistically between the intranasally and intradermally infected groups, as assessed by both the log-rank (Mantel–Cox) *p* = 0.3726 and the Gehan–Breslow–Wilcoxon tests *p* = 0.3291.

**Figure 2 viruses-16-00079-f002:**
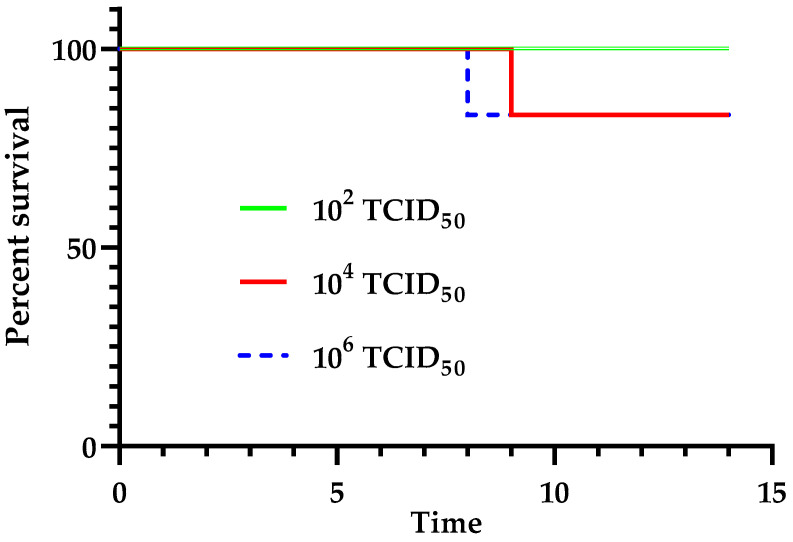
Kaplan–Meier survival curves of canaries infected with USUV via the intranasal route with 10^6^ TCID_50_ (*n* = 6), 10^4^ TCID_50_ (*n* = 6), or 10^2^ TCID_50_/canary (*n* = 6). The difference among the three groups was not statistically significant, as assessed by both the log-rank (Mantel–Cox) *p* = 0.5911, and the Gehan–Breslow–Wilcoxon tests *p* = 0.5888.

**Figure 3 viruses-16-00079-f003:**
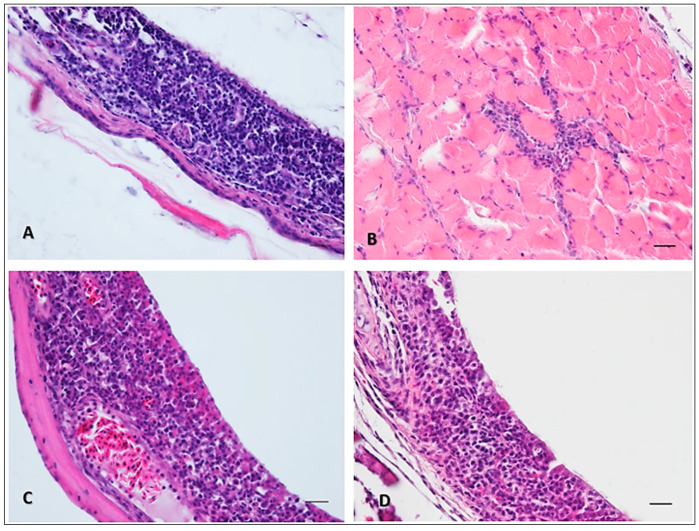
Pathological findings in two canaries lethally infected with 10^6^ TCID_50_ of USUV, by intradermal (**A**,**B**) or intranasal (**C**,**D**) inoculation. In both cases, a massive lymphoplasmacytic and histiocytic infiltration was observed in the lamina propria and submucosa, responsible for the thickening of the tissue, regardless of the inoculation site. Hematoxylin and eosin. Scale bars: 20 µm.

**Figure 4 viruses-16-00079-f004:**
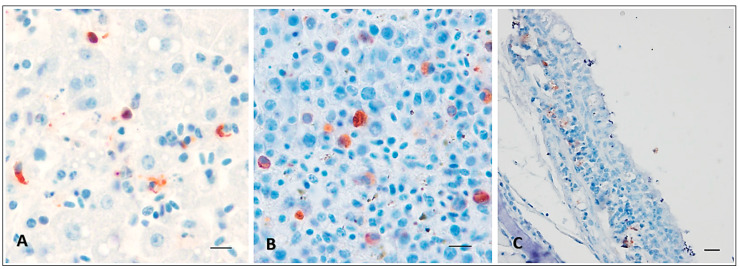
Immunohistochemical labeling of USUV antigens using a polyclonal rabbit anti-USUV antibody. Red-brown staining in antigen-positive cells in the liver (**A**) and spleen (**B**) from an intradermally infected canary dead at 6 dpi, but also in the trachea (**C**) of an intranasally infected canary (with 10^6^ TCID_50_) dead at 8 dpi. Gill’s hematoxylin counterstain. Scale bars: 5 µm (**A**,**B**), 10 µm (**C**).

**Figure 5 viruses-16-00079-f005:**
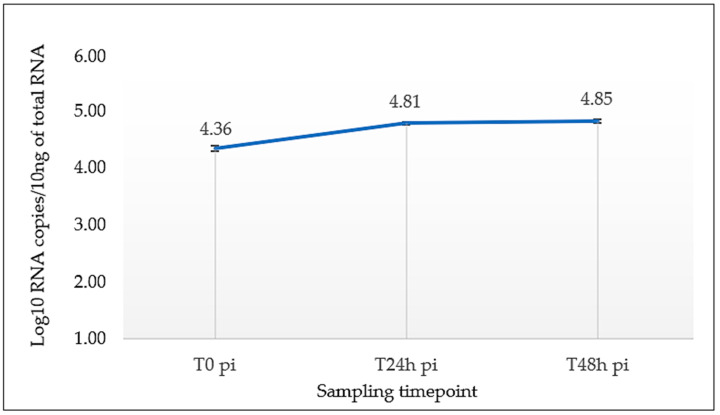
Viral RNA loads (Log10 RNA copies/10 ng of total RNA) detected in the culture supernatant of tracheal explants from canaries at 0, 24, and 48 h post-infection with USUV.

**Figure 6 viruses-16-00079-f006:**
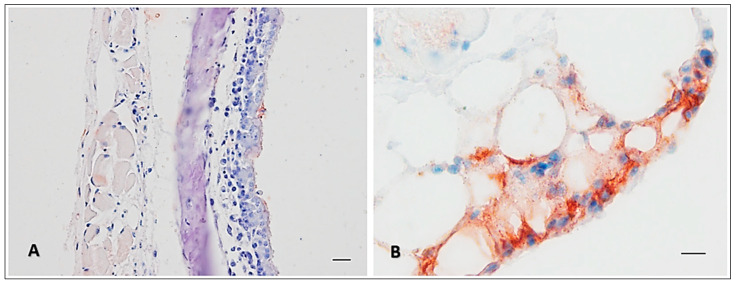
Immunohistochemical staining of USUV antigens using a polyclonal rabbit anti-USUV antibody. Red-brown staining in antigen-positive cells in the epithelial tracheal cells (**A**) and in cells from the surrounding adipose tissue (**B**). Gill’s hematoxylin. Scale bars: 10 µm (**A**) and 5 µm (**B**).

**Figure 7 viruses-16-00079-f007:**
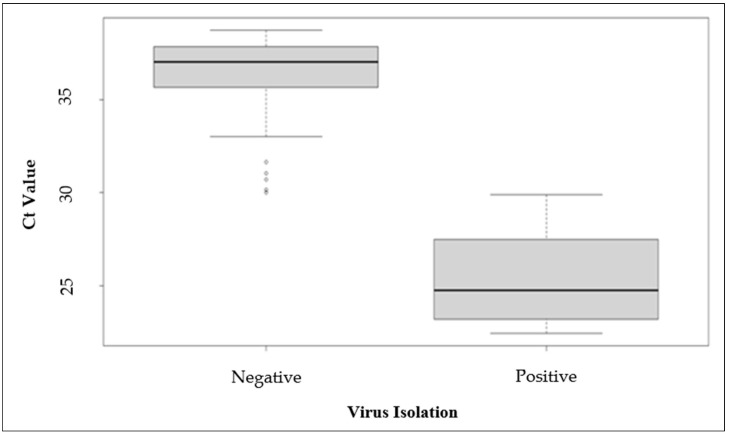
Box plot illustrating the possibility of isolating an infectious virus in cell culture depending on the Ct value of oral swabs determined by RT-qPCR for viral RNA.

**Figure 8 viruses-16-00079-f008:**
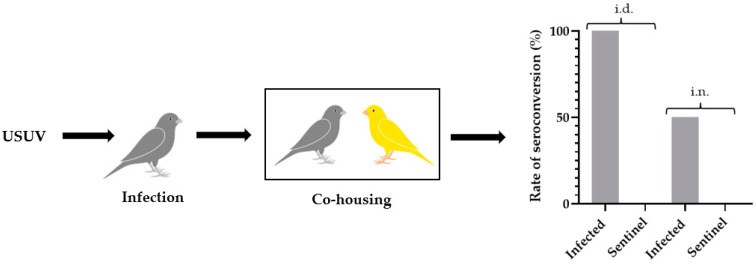
Schematic representation of the evaluation of the direct transmission of USUV infection to naive canaries. Canaries were infected intradermally or intranasally with 10^6^ TCID_50_/canary of USUV (i.d.-and i.n.-infected; grey; *n* = 6) and co-housed for 15 days with naive canaries (i.d. and i.n sentinels.; yellow; *n* = 6). Serum samples were collected 15 days post-infection, and an ELISA test was performed to evaluate the presence of antibodies against USUV.

**Figure 9 viruses-16-00079-f009:**
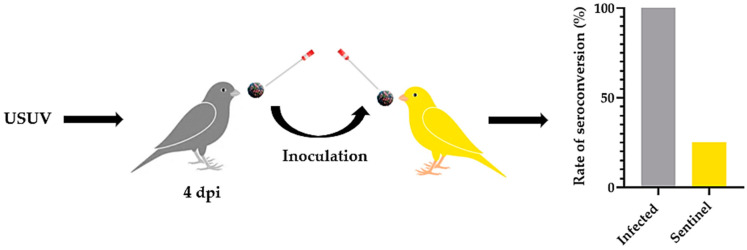
Schematic representation of the evaluation of the indirect transmission of USUV infection to naive canaries. Canaries were infected intradermally with 10^6^ TCID_50_/canary of USUV (infected; grey; *n* = 4), and an oral swab collected at 4 days post-infection (dpi) was inoculated in the oral cavity of naive canaries (sentinel; yellow; *n* = 4). Serum samples were collected 15 days post-infection, and an ELISA test was performed to evaluate the presence of antibodies against USUV. *p* = 0.071.

**Table 1 viruses-16-00079-t001:** Viral RNA loads (expressed as the number of USUV RNA copies/100 ng of total RNA) in the organs of lethally infected canaries, as determined by absolute quantification by RT-qPCR. “x” dpi corresponds to the day of death post-infection for each canary.

	Spleen	Liver	Brain	Heart	Lung	Kidney	Eye
Intranasal infection							
Canary 10^6^ TCID_50_ 8 dpi	8.78 ± 0.02	8.76 ± 0.02	6.45 ± 0.01	7.89 ± 0.01	8.84 ± 0.02	7.95 ± 0.02	8.06 ± 0.01
Canary 10^4^ TCID_50_ 9 dpi	6.06 ± 0.01	5.70 ± 0.01	4.57 ± 0.01	5.72 ± 0.02	6.90 ± 0.01	5.25 ± 0.01	6.92 ± 0.02
Intradermal infection							
Canary 6 dpi	8.75 ± 0.02	8.86 ± 0,01	5.78 ± 0.01	7.20 ± 0.02	8.21 ± 0.01	7.33 ± 0.03	8.21 ± 0.02
Canary 7 dpi	5.74 ± 0.03	5.01 ± 0.03	2.26 ± 0.04	4.37 ± 0.02	5.76 ± 0.01	3.41 ± 0.03	5.92 ± 0.02

## Data Availability

Data are contained within the article.

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
