# Peer review of "Evaluation of Non-Vector Transmission of Usutu Virus in Domestic Canaries (Serinus canaria)"

_viruses, 2024, doi:10.3390/v16010079_

Round 1
Reviewer 1 Report
Comments and Suggestions for Authors
Blanquer et al assess (in a timely manner) how USUV could over-winter. Given that USUV is an emerging threat, the topic is of relevance and comes in a timely manner. I have a few comments, especially on the statistical analyses that would need clarification, I believe. See attached.

Author Response
Dear Editor,
Please find in the attachement our point-by-point response to the Reviewers comments. Changes in the manuscript have been highlighted in yellow.
We wish to thank both Reviewers for their helpful and constructive comments.
Looking forward to hearing from you,
Prof. M. Garigliany

Reviewer 2 Report
Comments and Suggestions for Authors
Blanquer et al. evaluated bird-to-bird transmission of Usutu virus using a canary model. They found that Usutu virus is shed orally in inoculated birds, and intranasal inoculations of birds could lead to systemic infections. However, they found no evidence of direct bird-to-bird transmission, though they could infect birds at a low frequency using an oral swab from an infected bird. This suggests bird-to-bird transmission of Usutu virus is not likely to be a frequent event.
Major comments
1. Table 1 - was blood collected and tested for virus at the time of euthanasia? How do you explain the high levels of viral RNA in the brain, heart, lung but no antigen staining? Please discuss.
2. It would be helpful if the data described in lines 348-385 (% of birds shedding and RNA copies during shedding) was plotted in a graph. It is currently very hard to follow. Throughout the manuscript, rather than Cq values, viral RNA copies or genome equivalents should be presented.
3. Line 438 – 440 – I don’t think the authors have any evidence that the higher intranasal dose led to greater dissemination. Groups of 1 are presented in table 1. Please modify language.
Minor comments
1. It is unclear how many times the trachael explant experiment was performed (ie, trachae from how many birds).
2. In addition to comparing the viral dissemination data to previous experimentally infected birds, could the authors compare their data to previous data from naturally infected birds?
Comments on the Quality of English LanguageMinor editing for clarity is suggested.
Author Response

(The authors gave the same response as above.)
